# Novel Adipokines and Their Role in Bone Metabolism: A Narrative Review

**DOI:** 10.3390/biomedicines11020644

**Published:** 2023-02-20

**Authors:** Fnu Deepika, Siresha Bathina, Reina Armamento-Villareal

**Affiliations:** 1Division of Endocrinology, Diabetes and Metabolism, Department of Medicine, Baylor College of Medicine, Houston, TX 77030, USA; 2Center for Translational Research on Inflammatory Disease, Michael E DeBakey Veterans Affairs (VA) Medical Center, Houston, TX 77030, USA

**Keywords:** adipokines, bone metabolism, osteoblast, osteoclast, bone turnover markers, bone mineral density, osteoporosis

## Abstract

The growing burden of obesity and osteoporosis is a major public health concern. Emerging evidence of the role of adipokines on bone metabolism has led to the discovery of novel adipokines over the last decade. Obesity is recognized as a state of adipose tissue inflammation that adversely affects bone health. Adipokines secreted from white adipose tissue (WAT) and bone marrow adipose tissue (BMAT) exerts endocrine and paracrine effects on the survival and function of osteoblasts and osteoclasts. An increase in marrow fat is implicated in osteoporosis and, hence, it is crucial to understand the complex interplay between adipocytes and bone. The objective of this review is to summarize recent advances in our understanding of the role of different adipokines on bone metabolism. Methods: This is a comprehensive review of the literature available in PubMED and Cochrane databases, with an emphasis on the last five years using the keywords. Results: Leptin has shown some positive effects on bone metabolism; in contrast, both adiponectin and chemerin have consistently shown a negative association with BMD. No significant association was found between resistin and BMD. Novel adipokines such as visfatin, LCN-2, Nesfatin-1, RBP-4, apelin, and vaspin have shown bone-protective and osteoanabolic properties that could be translated into therapeutic targets. Conclusion: New evidence suggests the potential role of novel adipokines as biomarkers to predict osteoporosis risk, and as therapeutic targets for the treatment of osteoporosis.

## 1. Introduction

Adipose tissue secreted adipokines have both pro-inflammatory (leptin, resistin, RBP-4, lipocalin 2, NAMPT) and anti-inflammatory properties (omentin-1, adiponectin, SFRP5) that mediate pathophysiology of several bone diseases [1]. The more recent adipokines such as omentin-1 and vaspin not only have insulin-sensitizing properties and protective effects on cardiovascular health [2,3], but also have a significant role in bone metabolism. A few studies have found a positive association between vaspin and BMD serving as a protective biomarker against osteoporosis [3,4], while the effects of omentin-1 on BMD are conflicting [5,6]. The correlation between chemerin with BMD has been consistently negative [7,8,9], whereas no convincing evidence has been found on the association between visfatin [10,11] and lipocalin 2 [12,13] with BMD. A few other adipokines, such as RBP-4, have been shown to promote osteogenesis in vivo only, while adipokine apelin enhances human osteoblast proliferation both in vitro and in vivo [14,15,16]. In animals, nesfatin-1 may have a potential protective role on bone metabolism, however, evidence of a similar effect in humans is lacking [17,18].

Due to potentially conflicting findings between in-vitro and in-vivo studies, long-term research is warranted to establish a deeper understanding of the complex interplay between adipokines and bone. In the current review, we discuss the recent advances in our understanding of the role of adipokines, some of them novel, on bone metabolism.

## 2. Adipokines

### 2.1. Adiponectin

Adiponectin is one of the predominant adipokines that plays a crucial role in obesity, glucose, and lipid metabolism, as well as cardiovascular disease [19]. The role of adiponectin signaling in bone homeostasis remains complex, in part due to its multiple isoforms and receptor sub-types and conflicting results from animal versus human studies. Some of the recent advances made in the understanding of adiponectin are illustrated here.

#### 2.1.1. Pre-Clinical

A recent study sought to investigate the effect of adiponectin receptor activation (AdipoRon) on bone-fat balance and found that AdipoRon promoted osteoblastogenesis in pre-osteoblasts and inhibited osteoclastogenesis by inhibition of RANKL production in osteoblast. Moreover, mesenchymal stromal stem cells showed decreased adipogenesis when treated with AdipoRon [20]. In vivo studies have shown that adiponectin-deficient mice have reduced bone mass and increased adiposity [21]. Evidence from in vitro studies shows that adiponectin inhibits osteoclastogenesis by regulating RANKL/OPG ratio in the bone marrow microenvironment [21] and by suppressing nuclear factor-kB (NF-kB) and p38 signaling pathways, which are essential for osteoclast formation [22]. Over the last few years, calorie restriction (CR) to promote weight loss has gained attention when one study looking at the role of adiponectin on bone loss in mice subjected to short-term CR and found an increase in adiponectin expression with a decrease in BMD [23].

#### 2.1.2. Clinical Studies

Studies in postmenopausal women have shown both an san inverse relationship between adiponectin and BMD [24] and lower adiponectin among those with osteoporosis [25]. In a subset of patients with chronic kidney disease (CKD), adiponectin showed an inverse relationship with trabecular volumetric bone mineral density (vBMD), cortical vBMD, and cortical thickness [26]. Adiponectin also showed an inverse association with bone strength parameters in immobilized patients [27]. A recent case-control study further showed a strong inverse relationship between adiponectin and T scores in women with osteoporosis and osteopenia [28]. Results from a large prospective study showed that high adiponectin levels were associated with a greater risk of fractures in men independent of body composition and BMD but not in women suggesting that adiponectin may be a novel predictor of increased fracture risk but only in males [29].

In summary, the majority of the pre-clinical data suggests adiponectin has a positive influence on bone homeostasis via the reduction in osteoclast activity and increase osteoblastic differentiation whereas clinical studies showed a negative association between adiponectin with BMD and fractures.

### 2.2. Leptin 

Leptin is yet another adipokine that is involved in energy homeostasis and bone metabolism. Evidence from previous studies have demonstrated that leptin regulates bone mass by both peripheral and central mechanisms, although in vivo effects remain controversial to date [30]. Herein we summarize our current knowledge and understanding of leptin’s effect on bone metabolism.

#### 2.2.1. Pre-Clinical 

Data from studies over the last two decades have consistently shown that reduced leptin signaling in mice has differential effects on the axial and appendicular skeleton. The bone phenotype of leptin-deficient ob/ob mice shows an increase in trabecular bone volume (TBV), bone mineral content (BMC), and BMD in the spine and a decrease in femur length, BMC, and BMD in the limbs [31,32,33,34]. Central administration (intraventricular) of leptin in ob/ob mice has osteoclastic effects on spine TBV compared to osteoanabolic effects on cortical bone of the femur [31,32,35]. Leptin is also known to have direct effects on both osteoblasts and osteoclasts. Animal studies showed that leptin stimulates the proliferation of osteoblasts with no effect on mature osteoclasts in mice and subcutaneous administration of leptin reduced bone fragility [36]. In vitro studies show that leptin treatment results in bone mineralization and proliferation of human osteoblastic cells [37,38,39] while it has inhibitory effects on osteoclasts via suppression of the RANKL/OPG pathway [40,41]. Interestingly, studies have shown leptin expression during a unique time course of fracture healing in mice [42,43]. Philbrick et al. showed that long bone of ob/ob mice retain cartilage within growing bones and thus exhibit osteopetrotic phenotypes and that leptin gene therapy could potentially reverse osteopetrosis at the site of robust bone turnover in leptin-deficient (ob/ob) mice [44].

#### 2.2.2. Clinical Studies 

Most of the data on leptin and bone metabolism comes from studies conducted with women with hypothalamic amenorrhea (HA) as they have low circulating leptin levels. Two randomized controlled trials (RCTs) have shown contradictory findings on the effect of leptin administration on BMD in women with HA, one showing an increase in osteocalcin and a transient increase in urine N-terminal telopeptide (NTX), but no effects on BMD [45], while the other reported improvement in lumbar spine BMD in lean hypoleptinemic women [46]. A few years later, a study in young women with HA showed that 36 weeks of leptin treatment resulted in a decrease in parathyroid hormone (PTH) and RANKL/OPG ratio suggesting leptin decreases osteoclastic activity [47]. Studies in subjects with congenital lipodystrophy (CGL) who are leptin-deficient showed no change in bone mineral content (BMC) with leptin replacement [48]. Since weight loss is accompanied by both bone loss and a decline in leptin levels, the effects of leptin administration on bone metabolism following weight loss were evaluated in a small prospective study that showed a failure of leptin to prevent weight loss associated-bone loss [49].

In summary, the peripheral and central effects of leptin administration on bone remain controversial. Evidence from animal and human studies suggests anabolic effects of leptin on the bone, although association studies between leptin and BMD in humans have shown mixed results. Hence, large prospective longitudinal studies, including clinical trials, are needed to study the regulatory effect of leptin on bone and fracture risk.

### 2.3. Visfatin 

Visfatin, also known as pre-B cell colony-enhancing factor (PBEF) or nicotinamide phosphoribosyltransferase (NAMPT), is primarily synthesized by visceral adipose tissue [50]. NAMPT/visfatin is involved in energy metabolism and is also known to have pro-inflammatory properties [51] and, thus, may be helpful as a potential biomarker for obesity, insulin resistance, and type 2 diabetes (T2D) status [52].

#### 2.3.1. Pre-Clinical

Animal studies have shown that the knockdown of NAMPT in the mouse mesenchymal cell line C3H10T1/2 cells results in decreased Sirtuin1 activity and nicotinamide adenine dinucleotide NAD^+^ resulting in enhanced adipogenesis and reduced osteoblast differentiation in aged mice [53]. An earlier study investigating the role of visfatin in human osteoblasts showed that visfatin has insulinmimetic properties and that visfatin-mediated regulation of glucose uptake, proliferation, and type I collagen production in human osteoblasts involves insulin receptor (IR) phosphorylation, the same signal-transduction pathway used by insulin [54]]. Visfatin suppresses osteoclastogenesis by inhibiting the differentiation of osteoclast precursors into tartrate-resistant acid phosphatase (TRAP) positive multinucleated cells through the suppression of RANK, nuclear factor of activated T-cells c1 (NFATc1) and cathepsin-K [51]. However, visfatin inhibition of bone-resorption was seen only in the pre-osteoclasts rather than mature osteoclasts [55].

#### 2.3.2. Clinical Studies

Visaftin levels are known to increase with mechanical unloading thus adversely affecting bone health. A study of 12 healthy men showed that visfatin concentrations almost doubled after one day of microgravity and remained elevated even after recovery [56]. Studies have found no clear association between visfatin and BMD. One small study found a positive correlation between plasma visfatin and lumbar spine BMD in men [57], but none in post-menopausal women being treated for osteoporosis [10]. A large systematic review and meta-analysis of 59 studies found no convincing evidence of a relationship between visfatin and BMD [11].

In summary, in-vitro data suggest that visfatin influences bone metabolism by inhibiting osteoclast-mediated bone resorption primarily by targeting pre-osteoclasts. On the other hand, there is no convincing evidence of its association with BMD in human studies.

### 2.4. Chemerin 

Chemerin, also known as retinoic acid receptor responder 2 (Rarres2), or tazarotene-induced gene 2 (TIG2), was first described by Nagpal et al. in 1997 and was originally identified as a gene upregulated in psoriatic skin by synthetic retinoid [58]. Further studies revealed that chemerin is highly expressed in white adipose tissue (WAT), liver, and lung, while its receptor chemokine-like receptor 1 (CMKLR1) is predominantly expressed in adipocyte and immune cells [59,60]

In 2007, Bozaoglu et al. reported the level of chemerin and its receptor, chemokine-like receptor 1 (CMKLR1), were significantly high in individuals with obesity and T2D [59]. Several other studies have shown an association between elevation in chemerin expression and metabolic and inflammatory diseases including psoriasis, obesity, T2D metabolic syndrome, and cardiovascular disease [59,61]. Moreover, the dysregulated ratio of chemerin to adiponectin could play a role in metabolic syndrome [62]. However, its role as a regulator of bone mass needs further investigation.

#### 2.4.1. Pre-Clinical 

A study by Muruganandan et al. showed that knockout of chemerin or CMKLR1expression abrogated adipocyte differentiation, and proliferation of MSCs and increased osteoblast marker gene expression and mineralization in response to osteoblastogenic stimuli [11]. A few years later, the same authors provided further insight into mechanisms of lineage determination of MSCs and showed that chemerin/CMKLR1 influences changes in the expression of key adipogenic and osteoblastogenic transcription factors to limit osteoblastogenic Wnt signaling [63]. In vitro, chemerin induces bone resorption activity in mature osteoclasts via ERK5 phosphorylation. In vivo studies show that db/db mice exhibit increased levels of chemerin, CMKLR1, and cathepsin K mRNA expression and increased bone loss [64]. Treatment of db/db mice with CCX832 (a CMKLR1 antagonist) inhibits bone loss [64]. In a recent study, the knockdown of chemerin led to enhanced expression of genes involved in osteoblastogenesis and inhibition of genes in the osteoclastogenesis pathways in MC3T3-E1 and Raw264.7 cells, respectively [65]. In the same study, Rarres2^−/−^ mice had increased BMD, trabecular thickness, and bone formation marker bone-specific alkaline phosphatase (BALP), but decreased in both trabecular separation (Tb.Sp) and bone resorption marker TRAP-5b. [65]. There is data suggesting that CMKLR1 positively regulates bone remodeling in male mice in an androgen-dependent manner [66]. Accordingly, CMKLR1 knockout mice showed a decreased mRNA expression of testosterone synthesis enzymes in the testis as well as decreased mRNA expression of osteogenic markers and increased mRNA levels for osteoclast markers [66].

#### 2.4.2. Clinical Studies 

A case-control study in humans showed that chemerin levels are inversely correlated with BMD at both the lumbar spine and femoral neck [7]. Another study in obese Chinese post-menopausal women showed that serum chemerin levels were negatively correlated with BMD at the lumbar site after controlling for age, lean, and fat mass [9]. The results of a large population-based study in Pomerania showed that chemerin levels were inversely related to bone quality only in a sub-group of obese men and women, and not for those who were overweight and lean. The study also noted a positive association between chemerin levels and osteoporosis risk [8]. Thus, one can conclude that the effect of chemerin on bone quality is influenced by BMI.

In summary, altogether, the above data suggest that chemerin affects the remodeling cycle by inhibiting Wnt/β-catenin signaling resulting in reduced osteoblast differentiation and stimulating osteoclast differentiation and proliferation via activation of RANK signaling. Thus, chemerin could be useful as a potential biomarker for osteoporosis.

### 2.5. Omentin-1 

Omentin-1, also known as intelectin-1, was originally found in Paneth cells of the small intestine and is an intestinal lactoferrin receptor [67]. It is one of the novel adipocytokines that influences glucose metabolism, serving as a predictor for diabetes [68], and has anti-inflammatory [69], anti-atherosclerotic, and cardiovascular protective effects [70]. Some experimental evidence indicates that omentin-1 has anti-inflammatory, antioxidative, anti-apoptotic, and microbial defense roles [71]. Its role in bone metabolism is illustrated by the studies shown below.

#### 2.5.1. Pre-Clinical 

Omentin-1 has been shown to inhibit osteoblast differentiation and reduced osteoclast formation through stimulation OPG while inhibiting RANKL production in osteoblasts [37]. In vivo, in ovariectomized mice, omentin-1 ameliorated bone loss from estrogen deficiency by downregulating RANKL/OPG ratios [72]. Furthermore, another study showed that omentin-1 promotes osteoblasts differentiation via the TGF-β/Smad signaling pathway in MC3T3-E1by increased expression of runt-related transcription factor 2, collagen1, osteopontin, osteocalcin, and osterix [73]. Omentin-1 is also shown to impact fracture healing in studies where omentin-1 knockout (omentin-1^−/−^) mice had delayed fracture healing, accompanied by increased inflammation and osteoclast formation, and decreased angiogenesis [74].

#### 2.5.2. Clinical Studies 

Despite earlier studies showing that omentin-1 induces the proliferation of the human osteoblasts (HOB) pathway, human studies have shown that omentin-1 predominantly negatively impacts bone metabolism. A study in Iranian postmenopausal women showed an inverse correlation between omentin-1 and lumbar spine BMD [6] while another showed no significant correlation between omentin-1 and BMD at different skeletal sites [75]. Data from studies in Chinese women shows a negative correlation of omentin-1with bone-specific alkaline phosphatase (BAP) and NTX in premenopausal women [76] as well as a negative correlation between omentin-1 levels and BMD at different skeletal sites, irrespective of age, sex, and body mass index (BMI) [77]. This same study identified a cutoff value for omentin-1 of 15.37 ng/mL to predict osteoporosis in female subjects with diabetes with a sensitivity of 71.7%, and a specificity of 58.5% [77]. Likewise, a study found significantly higher omentin-1 in postmenopausal women with osteoporosis [78]. The negative association of omentin-1 with bone turnover markers (BTM)’s was also reported in women with anorexia nervosa (AN) [79]. On the other hand, a study in Chinese men found no correlation between omentin-1 and BMD at different skeletal sites and BTM [5].

In summary, pre-clinical data suggest the protective effects of omentin-1 on bone health. However, clinical studies reported a negative association between omentin-1 with BMD and higher levels of omentin-1 in women with osteoporosis. Due to these inconsistent findings, the potential usefulness of omentin-1 in osteoporosis prevention and therapy remains controversial.

### 2.6. Lipocalin-2 (LCN2) 

Lipocalin-2 (LCN2) is a novel osteoblast-derived adipokine consisting of 198 amino acids. It is also known as Neutrophil gelatinase-associated lipocalin (NGAL) [80]. Neilsen et al.1996 proposed that it might be a scavenger of bacterial products at sites of inflammation [81]. Recent advances in knowledge suggests that secretion of LCN2 is 10-fold higher in osteoblasts compared to white adipose tissue (WAT) as was previously understood [82]. Experiments in mice demonstrate that osteoblast-derived LCN2 maintains glucose homeostasis by inducing insulin secretion and improves glucose tolerance and insulin sensitivity. In addition, it regulates appetite suppression in a MC4R-dependent manner [82]. While the role of LCN2 in regulating gluco-metabolic parameters and the effect of metabolic dysregulation on LCN2 remain complex [83,84,85] evidence for its regulatory effects on bone metabolism is rapidly emerging.

#### 2.6.1. Pre-Clinical

Previous work in mice suggested up-regulation of LCN2 with mechanical unloading, and that LCN2 expression decreases osteoblast differentiation and promotes osteoclast activity [12]. A few years later, the same authors found that systemic ablation of LCN2 induced marked osteopenia, thus conflicting with the prior observations [13]. In this study, Lcn2^−/−^ mice exhibited lower TBV, lesser trabecular number, and higher Tb.Sp when compared to wild-type (WT)mice. Lcn2^−/−^ mice showed a lower osteoblast number over a bone surface, and subsequently a significantly lower bone formation rate, while osteoclast variables were unremarkable [13]. These authors also reported that the effect of LCN2 on bone is mediated by alteration in glucose transport protein GLUT 1which is essential for osteogenesis. In contrast, Mosailou et al. showed that LCN2-deficient mice had normal bone mass and concluded that LCN2 does not have an a-effect on bone [82].

#### 2.6.2. Clinical Studies

A prospective study in a cohort of elderly women demonstrated that high levels of circulating LCN2 predict the future risk of osteoporotic fractures requiring hospitalization [86]. In contrast, a few studies found no association between LCN2 and BMD in postmenopausal women with osteoporosis [87,88,89]. Interestingly, however, there is new evidence of the potential link between LCN2 and the progression of chronic kidney disease -mineral bone disorder (CKD-MBD) as well as catabolic effects of LCN2 in OA joints [90,91]

In summary, mice lacking LCN2 show poor bone microarchitecture. In addition, LCN2 levels are increased in response to unloading suggesting that it may mediate the development of osteoporosis from disuse. However, clinical studies show no convincing evidence of its association with BMD in postmenopausal women with osteoporosis.

### 2.7. Resistin

Resistin is an adipocyte-derived cytokine that has been linked to obesity, diabetes, insulin resistance, and atherosclerosis [92,93,94,95]. Resistin is also known to regulate the production of pro-inflammatory cytokines [96] and plays a role in energy homeostasis [97].

#### 2.7.1. Pre-Clinical 

Resistin was found to have no effect on the differentiation of bone marrow MSC towards adipogenesis or osteogenesis [98]. One of earlier studies found that resistin is expressed in both pre and mature osteoblasts as well as pre-osteoclasts and is upregulated during the differentiation of both osteoclasts and osteoblasts [99]. An in-vitro study showed that resistin mRNA expression increased, and glucose uptake decreased in RAW264.7 cells treated with RANKL [100] suggesting that osteoclasts could mediate insulin sensitivity through resistin secretion.

#### 2.7.2. Clinical Studies 

Several studies have evaluated the relationship between resistin and BMD in both men and post-menopausal women and found no significant association [89,101,102,103]. Similarly, a large systematic review and meta-analysis showed no association between resistin levels and BMD [11]. A weak negative association between resistin and lumbar spine BMD was seen in a study involving middle-aged men [104]. However, the results of a recent study showed that post-menopausal women with osteoporosis had higher serum resistin levels and low vitamin D with vitamin D as an independent predictor of serum resistin levels [105]. Interestingly, obesity does not affect the secretion of resistin in obese women with postmenopausal osteoporosis [106], however, combined aerobic and resistance exercises decrease resistin levels [107].

In summary, studies have shown no effect of resistin on BMD. Although resistin appears as a significant link between obesity and T2D, it has no significant role in regulating bone metabolism.

### 2.8. Nesfatin-1

In 2006, Oh-1 and colleagues identified a novel anorexigenic adipokine nesfatin, also known as non-esterified fatty acid/nucleobindin2 (NEFA/NUCB2) produced in the hypothalamus and released as a fragment, nesfatin-1 into the circulation. Nesfatin-1 is located in the N terminus of NUCB2 and is indispensable to the induction of satiety in vivo [108]. Studies have shown high serum levels of nesfatin-1 in patients with OA [109]. In vitro nesfatin-1 induces the production of pro-inflammatory agents, such as COX-2, IL-8, IL-6, and MIP-1α in human primary chondrocytes from OA patients [110]. While animal studies have shown the anti-inflammatory and anti-apoptotic effects of nesfatin-1 on rat chondrocytes and potential effects on ameliorating OA [111], currently there is limited understanding of the role of nesfatin-1 on bone metabolism [112].

#### 2.8.1. Preclinical

Recent evidence from animal studies has shown that nesfatin-1 regulates bone metabolism and could limit bone loss. Nesfatin-1 treated ovariectomized female rats with established osteopenia showed preservation of bone architecture, and increased bone strength [17]. A similar study in ovariectomized rats showed that administration of nesfatin led to a significant increase in serum osteocalcin and BAP and TvBMD and TbvBMD by pQCT in the metaphysis of long bones compared to the control group [112]. A study evaluating the effect of mechanical loading on rat humeri showed that bone strength parameters measured under various mechanical loading conditions increased after the nesfatin-1 administration [18].

#### 2.8.2. Clinical

To date, there are no other clinical studies investigating the effects of nesfatin-1 on bone.

Conclusion: Although pre-clinical evidence points towards a bone-protective effect of nesfatin-1, clinical studies supporting its influence on bone metabolism are currently lacking. Since nesfatin-1 acts as an anorexigenic adipokine and pre-clinical evidence suggests its role in preserving bone density, it could serve as a promising target for the treatment of both obesity as well as osteoporosis.

### 2.9. RBP-4 

RBP-4 is an adipocyte-secreted molecule that is elevated in the serum before the development of frank diabetes [113]. RBP-4 is correlated with the magnitude of insulin resistance independent of obesity [113], however, the effect of RBP-4 on bone metabolism is poorly understood.

#### 2.9.1. Preclinical

Animal data have implicated the involvement of chondrocytic RBP-4 in bone growth, particularly in the formation of the secondary ossification centers of the limb [114]. In vitro, retinol has an inhibitory effect on bone, whereas, in vivo, mice fed retinol showed increased expression of osteogenic genes with little effect on BMD [115].

#### 2.9.2. Clinical

While one study showed that levels of RBP-4 may positively correlate with bone resorption marker CTX in males with T2D [116], another found a positive relationship between RBP-4 and BMD at different sites in postmenopausal women with osteopenia/osteoporosis [10,117]. However, earlier studies in young men did not find any significant association between RBP-4 and BMD [118].

In summary, current animal data suggest possible osteoanabolic effects of RBP-4 on bone homeostasis, however, its role in human skeletal health remains undefined.

### 2.10. Apelin

Apelin is a recently discovered peptide that is the endogenous ligand for the orphan G-protein-coupled receptor APJ. Apelin and APJ are expressed in a variety of tissues with predominant protective effects on cardiac vasculature [119]. Animal studies have revealed exciting development in the treatment of obesity and diabetes using apelin analogs [120]. Apelin has also been implicated in the regulation of skeletal homeostasis as well as in the pathogenesis of osteoarthritis (OA).

#### 2.10.1. Pre-Clinical 

Previous studies found apelin expression on human osteoblasts enhancing osteoblast proliferation and inhibiting osteoblast apoptosis via APJ/PI3 kinase/Akt pathway but had no effect on osteoblast differentiation [14,15,16]. A few years later, a group of investigators reported that apelin gene knockout (APKO) mice had increased bone mass compared to wild-type mice, thus, suggesting the anti-anabolic effects of apelin in vivo [121]. In contrast, a more recent study demonstrated that apelin was down-regulated in rats with ovariectomy-induced osteoporosis, and treatment with apelin-12 restored bone mass and microstructure [122]. Furthermore, apelin-deficient mice were observed to exhibit decreased total area and decreased periosteal and endocortical bone surfaces [123]. Importantly, a study using a rat tibial osteotomy model showed that local injection of apelin protein improved bone healing based on imaging and histological analyses which has implications for its potential use in fracture healing [124]. Evidence from both in-vitro and in-vivo studies suggests apelin promotes osteogenic differentiation of hMSCs partly via the Wnt/β-catenin signaling pathway [123,124].

#### 2.10.2. Clinical Studies 

A study reported decreased expression of both apelin-13 and vitamin D3 in osteoporosis patients versus healthy controls [123]. Likewise, another study demonstrated significantly lower levels of apelin-13 in subjects compared to those without osteoporosis [125]. In this study, apelin-13 level correlated positively with BMD and P1NP, but negatively with age and CTX [125].

In summary, animal studies have shown that apelin regulates bone metabolism by enhancing osteoblastogenesis and inhibiting osteoblast apoptosis. A few observational studies in humans support the positive effect of apelin on BMD, though more studies are needed to determine its utility as a potential therapeutic target for osteoporosis.

### 2.11. Vaspin 

Vaspin is a novel adipokine derived from visceral adipose tissue identified as a member of the serine protease inhibitor family that exerts insulin-sensitizing effects in obesity [2]. Vaspin has protective effects on endothelial progenitors and anti-atherogenic properties [3]. Evidence suggests that it could potentially ameliorate vascular complications from hyperglycemia in patients with T2D [3]. However, knowledge of its role in regulating bone metabolism is limited but rapidly emerging.

#### 2.11.1. Pre-Clinical 

Earlier studies investigating the effect of vaspin on human osteoblast showed that vaspin inhibits osteoblast apoptosis through upregulation of Bcl-2 and downregulation of Bax through activation of the MAPK/ERK signaling pathway [126]. Vaspin has a regulatory effect on bone marrow stem cells (BMSCs) as shown by the results of a study where vaspin treatment at a concentration of 150 ng/mL promoted the proliferation of BMSCs probably by activation of the PI3K/AKT signaling pathway [127]. Moreover, vaspin has been suggested to downregulate RANKL-induced osteoclastogenesis by inhibiting the expression of transcription factor NFATc1 in RAW264.7 cells and bone marrow cells [128] with protective effects on osteoblast differentiation [129,130]. In contrast, one study showed that vaspin could inhibit osteogenic differentiation in vitro, through increased expression of miRNA-34c and activation of the PI3K-Akt signaling pathway [131]. In vivo, mice fed a high-fat diet (HFD) when treated with vaspin showed reduced body weight, enhanced bone strength, trabecular bone mass, and expression of Runx2, Osx, PINP, and decreased plasma CTX. Vaspin also upregulated mRNA expression of osteogenesis-related genes Runx2, Osx, and Colla1 and protein expression of Runx2, Smad2/3, and p-Smad2/3 [130].

#### 2.11.2. Clinical Studies

Human studies on vaspin and bone health have shown mixed results. A small study evaluated the relationship between vaspin and bone indices in girls with AN and found significantly higher levels of vaspin which negatively correlate with the OPG/RANKL ratio in subjects with AN compared to healthy controls [132]. A study in patients with inflammatory bowel disease (IBD) showed no significant association between vaspin and BMD but chemerin and visfatin were associated with the development of osteoporosis in these patients [133]. However, a cross-sectional study in post-menopausal women showed a positive association between vaspin and BMD at the femoral neck and total hip [4]. Vaspin levels were also found to increase after 6 months of treatment with bisphosphonates in post-menopausal women. [134]. In a study involving Iranian women, those with the lower resting metabolic rate (RMR) had higher BMI and BMD, and the only adipokine mediating the link between the RMR and BMD was omentin-1 but not vaspin [135].

In summary, vaspin is a novel adipokine that promotes osteoblast proliferation and inhibits osteoclastogenesis in vitro. Additional studies to fully elucidate its role in human bone homeostasis will determine its future utility in the therapy and monitoring response to treatment in patients with osteoporosis.

## 3. Cross Talk between Adipokines and Bone

### 3.1. Transcriptional Factors That Modulate Adipocytes and Subsequent Effect on Osteoblastogenesis

The fate of differentiation of bone marrow MSCs, to either adipogenic or osteoblastogenic programs, plays an important role in the regulation of bone homeostasis [136]. Studies confirmed that extracellular signaling proteins, including the adipocyte-specific peroxisome proliferator-activated receptor gamma (PPARγ) or myogenic osteoblast initiation (inhibition of PPARγ or activation of Runx2/osterix) transcription factors, can alter the differentiation of osteoblasts [137,138,139]. Apart from Wnt1 signaling, several BMPs also stimulate osteogenesis [140]. For example, activation of type1A BMP receptor (BMPR-1A) induces PPARγ expression and subsequent adipocyte differentiation [141]. On the contrary, activation of the type IB BMP receptor (BMPR-IB) induces Runx2 expression and favors osteoblastogenesis [142]. IGF1 along with insulin was reported to activate the signaling cascade of differentiation of MSCs to adipogenic or osteoblastogenic pathways in patients with obesity and diabetes [143]. The crucial shift of MSC differentiation to adipogenic rather than osteogenic lineage is linked to reduced bone formation observed in both obese and diabetic patients. Of all factors, PPARγ remains the key driver of adipocyte differentiation by binding to the c-fos promoter of hematopoietic stem cells (HSCs), thereby abrogating myogenic and stimulating osteoclastogenic cascade. Consistent with these findings, selective deletion of osteoclastogenic PPARγ has been found to increase bone volume from reduced bone resorption [144]. In addition, the collagen composition of the bone matrix is also believed to have an influence on MSC differentiation to myogenic lineage. This was confirmed by a study of cultured MSCs on denatured collagen which resulted in the promotion of adipogenesis [145]. Thus, as illustrated in Figure 1, a shift in MSC differentiation to adipogenesis may be linked to PPARγ and other transcriptional factors leading to excessive or abnormal bone resorption. The cross-talk between bone forming osteoblasts and osteoclasts, can be decoupled by number of pro-adipogenic factors leading to disruption of normal homeostasis of bone. This tight regulation might be lost/enhanced by the several adipokine derivatives that can activate osteoclastogenesis in PPARγ-dependent manner.

### 3.2. Role of Adipokine Machinery on Osteoblast Differentiation and Bone Resorption

Apart from adipocyte differentiation, in the presence of PPARγ, the release of signaling proteins NFAT2/RANKL/Cathepsin can also affect bone homeostasis [146]. Adiponectin, the primary adipokine, has been reported to have a direct negative effect on MSC osteoblastogenesis [147]. Recently, this finding was supported by a study in aging and postmenopausal women have confirmed an inverse relationship between serum adiponectin levels and BMD [148]. On the contrary, the mode of action of Leptin remains unclear, whereas, in some studies, leptin acts directly on precursor cells to promote osteoblast differentiation and proliferation but in some studies, leptin was linked to reduced bone mass [31,149,150]. Chemerin, a novel adipokine with its receptor CMKLR1, has direct interaction with PPARc and promotes MSC adipogenesis [151]. In some other studies, simultaneous knock-out studies of CMKLR1, increased osteoblastogenesis by upregulation of osterix and suppressed osteoclast differentiation [63,79,152], but overall, the role of chemerin in relation to bone homeostasis remains unclear.

Similarly, as demonstrated in Figure 2, vaspin interacts with Grp78 a glucose-regulated protein, to induce intracellular signaling (PI3K/AMPK pathway) in vascular smooth muscle cells that not only improves glucose tolerance and insulin sensitivity but also enhance osteoblastogenesis in rats. [153]. Novel adipokines of apelin with receptor APJ [154], (angiotensin II receptor like-1), vaspin, and nesfatin-1, when bound to G-protein, coupled receptors (GPCRs) and cannabinoid receptors (CR), are known to maintain glucose tolerance [155,156] Additionally, in some studies of obesity, vaspin levels are low along with suppression of leptin and TNF-α synthesis [157].

However, like other adipokines, the overall functional aspect of vaspin is unknown. On the other hand, LCN2 is a glycoprotein binding to the MC4R receptor and is involved in iron homeostasis. LCN2 is abundantly expressed in WAT and is induced by inflammatory stimuli by activation of NF-Kb [158]. Serum concentrations of this protein are positively associated with adiposity, insulin resistance hyperglycemia, insulin resistance, and CRP levels [159]. LCN2 binds to MC4R to initiate the p38-MAPK pathway and results in the activation of osteoclast precursors [160]. Even though most of the adipokines initiate osteoblast activation and subsequent bone formation, the perspective and role of various novel adipokines on bone metabolism need to be re-examined to better understand the pathophysiology of bone.

## 4. Discussion

In this review, we summarized recent advances made in understanding the regulatory role of adipokines in bone metabolism. There is a wealth of literature and much ongoing research to help identify adipokines that could serve as potential targets for the prevention or treatment of osteoporosis. However, the incongruent results between pre-clinical and clinical data continue to be a challenge in interpreting and subsequently translating these findings into clinical practice. Given these are newer adipokines, there is limited data on human subjects.

Adiponectin has consistently shown a negative association with BMD, however, targeting low levels of adiponectin is debatable as it serves as a protective biomarker for other metabolic conditions such as insulin resistance and diabetes. Despite the positive effects of Leptin on bone metabolism as demonstrated by earlier studies, its therapeutic role is limited to metabolic conditions eg. Lipodystrophy, and hypothalamic amenorrhea, however, its role in metabolic bone disease remains undefined. Meanwhile, adipokine resistin though present in the osteoblasts and the osteoclasts has not been found to have any significant impact on BMD. Fortunately in the past few years, novel adipokines have been identified such as visfatin, LCN2, nesfatin-1, RBP-4, apelin, and vaspin that may have osteoprotective/osteoanabolic effects and thus could serve as potential therapeutic targets for osteoporosis and other metabolic bone diseases.

## Figures and Tables

**Figure 1 biomedicines-11-00644-f001:**
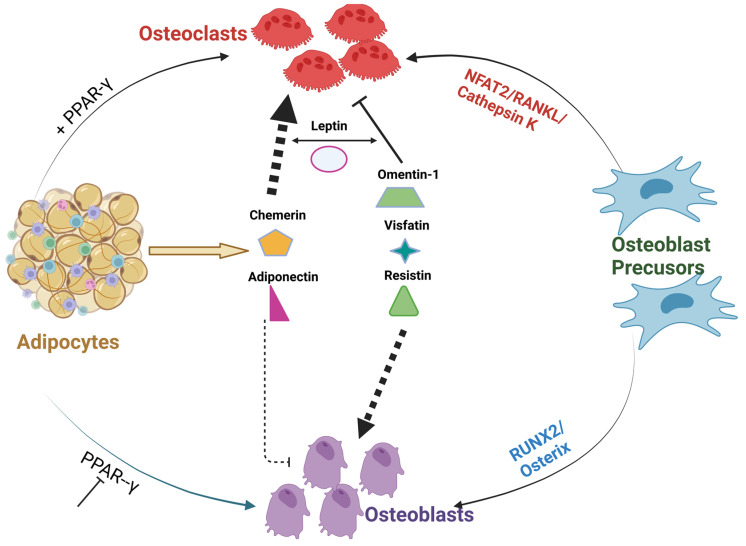
**Cross-talk between different Adipocyte derived factors and bone.** The above figure illustrates PPARγ inhibition or activation along with adipocyte-derived adipokines can alter or disrupt the communication between osteoblasts and osteoclasts. Adipokines; omentin visfatin and resistin enhance osteoblast differentiation and inhibit osteoclast formation. On the contrary, adipokines; leptin, chemerin, and adiponectin abrogate osteoblast formation and, thus, resulting in alterations in the homeostasis of bone [142,143,144].

**Figure 2 biomedicines-11-00644-f002:**
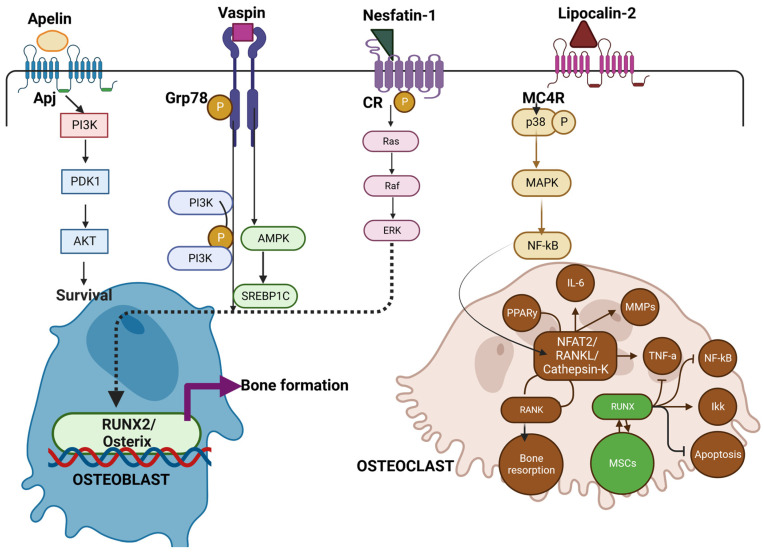
**Illustrates the role of Novel adipokines role in bone metabolism.** Apelin, vaspin, and Nesfatin when bound to receptors of APJ, Grp78, and Cannabinoid receptors (CR) activate Pi3k -Akt pathway and Ras pathway for subsequent osteoblast survival and RUNX2/Osterix pathway leading to bone formation. On the other hand, lipocalin (LCN2) binds to the MC4R receptor complex on OCPs, inducing the p38 MAP kinase pathway, then binds to Sp1 sites on the RANK promoter, activating RANK expression. This in turn favors osteoclast formation and bone resorption. Thus, novel adipokines mediating the crosstalk between osteoblast and osteoclast add on a new mechanism by which Osteoblast can control bone resorption [147,148,149,150].

## Data Availability

No new data were created or analyzed in this study. Data sharing is not applicable to this article.

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
