# Peer review of "Novel Adipokines and Their Role in Bone Metabolism: A Narrative Review"

_biomedicines, 2023, doi:10.3390/biomedicines11020644_

Round 1
Reviewer 1 Report
The Deepika et al., 2022, manuscript ID biomedicines-2133400 addresses the role of Adipokines in the Bone Metabolism and associated diseases.
There are few queries and few suggestions which makes this manuscript more representable to be publish.
1. The abstract need to be restructured and do not use the citations in the abstract?
2. A previous report showed that pre-treatment of testosterone is associated with greater improvement in BMD. Also the literature suggest the enhancement of testosterone in the adiponectin treated older individual. Did you find the literature having increased BMD in adiponectin treated old patients?
3. The authors must cite the contrary role of APN/Chemerin in recent articles (PMID: 29669464).
Author Response
- The Abstract has been restructured and citations have been removed
- That is a very interesting question and definitely worth exploring. We found one study in mice (PMID: 30471430) that showed exogenous treatment of adiponectin to aged mice resulted in marked improvements in testicular mass, histological features (cells proliferation), insulin receptor expression, testicular glucose uptake, anti-oxidative enzymes activity, and testosterone synthesis as compared with the control. However, we could not find any study that evaluated the effects of adiponectin treatment on BMD in elderly patients.
- Thank you for your suggestion. This literature has been referenced in Line 171. Reference 62.
Reviewer 2 Report
Thank you for inviting me to review the following manuscript on such an important and necessary topic. The investigators conducted a literature review in reference to the role of novel adipokines in bone metabolism.
The manuscript presents serious methodological errors and does not follow a structure according to the quality standards of the journal, being considered a scientific review of the literature is considered essential to follow a standard, we recommend the recommendations of PRISMA.
Comment 1
The work requires an English review by an expert; please attach certification of the English review.
Comment 2
Attach a checklist for review studies following the PRISMA model. Follow PRISMA guidelines.
Comment 3
Reformulate and rewrite the abstract with the following sections, they should be written in the form of: Background. This is a concise statement of the reasons for conducting this research, placing it in the context of current knowledge or controversies. Objective. This is a clear statement of the precise objective or question being addressed in the paper this may take the form of objectives or contrasting hypotheses. Methods. The basic design of the study and its duration should be described for quantitative and qualitative research quantitative and qualitative the methods used should be indicated and the data/statistical methods should be provided. Results. The main results of the study should be stated in narrative form any measurements or other information that may require explanation information that may require explanation confidence intervals are preferred to p-values values confounders, modifiers or mediators, as well as any other factors crucial to the outcome of the study should be indicated crucial to the outcome of the study should be indicated. Conclusions. The conclusions of the study that are directly supported by the evidence presented should of the evidence reported, along with clinical application, and speculation about the potential impact on current thinking current thinking.
Comment 4
Specify the type of study in the title of the manuscript.
Comment 5
Bibliographic references are not usually indicated in the abstract.
Comment 6
Provide a PROSPERO registration number and enter a subsection in the methodology called "protocol and registry".
Comment 7
The article has serious errors in the methodology, it is not known how the search for the articles that make up the manuscript was carried out; criteria for inclusion and exclusion of the included studies; evaluation of the risk of bias.
Comment 8
They should introduce a discussion section. Discussions should cover the key findings of the study: discuss any previous research related to the topic to place the novelty of the discovery in the appropriate context, discuss possible shortcomings and limitations in its interpretations, discuss its integration into the current understanding of the problem and how. This advances current views, speculates on the future direction of research, and freely postulates theories that could be tested in the future, completed, and reformulated. The discussion should be rewritten to present serious errors.
Comment 9
The order of the bibliographic references does not have a sequential order of appearance and the references have errors, please review the rules of the journal.
Author Response
Comment 1: We don't think that we need an English editor as all of us are fluent in written and conversational English.
Comment 2: We would like to clarify that this is a narrative review and not a systematic review. We now specify this in the Title.
Comment 3: We have restructured the abstract as suggested.
Comment 4: Done
Comment 5: We have removed the citations in the abstract.
Comment 6: We are not registered with PROSPERO as this is not a systematic review
Comment 7- This is a narrative review and not a systematic review. This is a comprehensive review of literature available in PubMed and Cochrane databases with emphasis on the last five years using the keywords: adipokines; bone metabolism; osteoblast; osteoclast; bone turnover markers; bone mineral density; osteoporosis.
Comment 8: We revised the discussion. Please see Line 517-525
Comment 9: The references appear in sequential order in the text of the manuscript. We used the software EndNote to key in the references.
Reviewer 3 Report
Novel Adipokines and their role in Bone Metabolism
Main observations:
The manuscript topic is consistent with the journal content.
The authors concluded, among other things, that 'The complex interplay between bone and adipose tissue is critical to an understanding of obesity and its impact on bone health'.
LACK of LIMITATION of review-analysis (at the end of the Discussion section).
Lack of precision in the name of adipokine and other biological substances in the manuscript body and Figure 1 and Figure 2.
Literature needs to be updated - more than 39% are articles more than ten years old - more current items should be used.
The central part of the discussion is consistent with the evidence and arguments and addresses the stated primary objective.
I believe this study would be a candidate for publication in your journal as an original article with minor revisions.
Minor observations:
Lack of EXPLANATION OF ABBREVIATIONS IN the MANUSCRIPT BODY UNDER FIGURE 1 AND FIGURE 2.
In Figure 1.
NO PRECISION in the names included in the graphic. Appears 'Chimerin' instead of 'Chemerin; Appears' PPARg' instead of 'PPARgamma;, Appears 'omentin' instead of 'omentin-1'.
In Figure 2
NO PRECISION in the names included in the graphic. Occurs 'PPARg' instead of 'PPARgamma', Occurs 'NF-Kb' instead of 'NF-kB'.
Selected examples of lack of precision are provided below.
Instead of:
LCN2 is abundantly expressed in WAT and is induced by inflammatory stimuli through activation of NF-Kb [157].
Should be:
LCN2 is abundantly expressed in WAT and is induced by inflammatory stimuli by activating NF-Kb [157].
Instead of:
It is one of the novel adipocytokine that influences glucose metabolism serving as predictor for diabetes [67], and has anti-inflammatory [68], anti-atherosclerotic, and cardiovascular protective effects [69].
Should be:
It is one of the novel adipocytokines that influences glucose metabolism serving as a predictor for type 2 diabetes [67], and has anti-inflammatory [68], anti-atherosclerotic, and cardiovascular protective effects [69].
Instead of:
The correlation between chemerin with BMD has been consistently negative [7-9] whereas no convincing evidence has been found on the association between visfatin [10, 11] and Lipocalin 2 [12, 13] with BMD.
Should be:
The correlation between chemerin with BMD has been consistently negative [7-9], whereas no convincing evidence has been found on the association between visfatin [10, 11] and lipocalin 2 [12, 13] with BMD.
Instead of:
NAMPT /visfatin is involved in energy metabolism and also known to have pro- inflammatory properties [51], thus, may be useful as potential biomarker for obesity, insulin resistance and Type 2 diabetes (T2D) status [52].
Should be:
NAMPT/visfatin is involved in energy metabolism and is also known to have pro-inflammatory properties [51]; thus, it may be helpful as a potential biomarker for obesity, insulin resistance and type 2 diabetes (T2D) status [52].
Instead of:
In 2007, Bozaoglu et al. reported the level of chemerin and its receptor, chemokine-like receptor 1 (CMKLR1), were significantly high in individuals with obesity and Type 2 diabetes [59].
Should be:
In 2007, Bozaoglu et al. reported that the level of chemerin and its receptor, chemokine-like receptor 1 (CMKLR1), were significantly high in individuals with obesity and type 2 diabetes [59].
Instead of:
It is also known as Neutrophil gelatinase-associated lipocalin (NGAL) [79].
Should be:
It is also known as Neutrophil gelatinase-associated lipocalin (NGAL) [79].
Instead of:
In 2006 Oh-1 and colleagues identified a novel anorexigenic adipokine nesfatin also known as non-esterified fatty acid/nucleobindin2 (NEFA/NUCB2) produced in the hypothalamus and released as a fragment, nesfatin-1 into the circulaion.
Should be:
In 2006 Oh and colleagues identified a novel anorexigenic adipokine nesfatin, also known as non-esterified fatty acid/nucleobindin2 (NEFA/NUCB2) produced in the hypothalamus and released as a fragment, nesfatin-1 into the circulation.
Instead of:
A similar study in ovariectomized rats showed that administration of nesfatin led to a significant increase in serum osteocalcin and BAP andTvBMD and TbvBMD by pQCT in the metaphysis of long bones compared to the control group [111].
Should be:
A similar study in ovariectomized rats showed that administration of nesfatin-1 led to a significant increase in serum osteocalcin and BAP and TvBMD and TbvBMD by pQCT in the metaphysis of long bones compared to the control group [111].
Instead of:
In contrast, a more recent study demonstrated that Apelin was down-regulated in rats with ovariectomy-induced osteoporosis and treatment with Aplein-12 restored bone mass and microstructure [121].
Should be:
In contrast, a more recent study demonstrated that apelin was down-regulated in rats with ovariectomy-induced osteoporosis and treatment with apelin-12 restored bone mass and microstructure [121].
Instead of:
There is evidence suggesting that it could potentially ameliorate vascular complications from hyperglycemia in patients with diabetes [3].
Should be:
Evidence suggests that it could potentially ameliorate vascular complications from hyperglycemia in patients with type 2 diabetes [3].
Instead of:
Studies confirmed that extracellular signaling proteins including the adipocyte specific (PPARg) or myogenic osteoblast initiation (inhibition of PPARg or activation of Runx2/osterix) transcription factors can alter the differentiation of osteoblasts[136-138].
Should be:
Studies confirmed that extracellular signaling proteins, including the adipocyte-specific (PPARgamma) or myogenic osteoblast initiation (inhibition of PPARgamma or activation of Runx2/osterix) transcription factors, can alter the differentiation of osteoblasts [136-138].
Instead of:
Chimerin, a novel adipokine with its receptor CMKLR1, have direct interaction with PPARc and promoted MSC adipogenesis [150].
Should be:
Chemerin, a novel adipokine with its receptor CMKLR1, have direct interaction with PPARc and promotes MSC adipogenesis [150].
Author Response
Main observations:
LACK of LIMITATION of review-analysis (at the end of the Discussion section). Response: Given these are newer adipokines, there is limited data on human subjects. Please see Line 513.
Lack of precision in the name of adipokine and other biological substances in the manuscript body and Figure 1 and Figure 2. Response: Both Figures 1 and 2 are revised and the name of adipokines is now consistent throughout the manuscript.
Literature needs to be updated - more than 39% are articles more than ten years old - more current items should be used. Response: The older articles have been included as background to improve discussion on the newer adipokines but the majority of the literature used in the manuscript are published in the last 5 years.
Minor observations:
Lack of EXPLANATION OF ABBREVIATIONS IN the MANUSCRIPT BODY UNDER FIGURE 1 AND FIGURE 2. Revised according to the reviewer's suggestion
In Figure 1.
NO PRECISION in the names included in the graphic. Appears 'Chimerin' instead of 'Chemerin; Appears' PPARg' instead of 'PPARgamma;, Appears 'omentin' instead of 'omentin-1'. Revised according to reviewer's suggestion
In Figure 2
NO PRECISION in the names included in the graphic. Occurs 'PPARg' instead of 'PPARgamma', Occurs 'NF-Kb' instead of 'NF-kB'. Revised according to reviewer's suggestion
Selected examples of lack of precision are provided below.
Instead of:
LCN2 is abundantly expressed in WAT and is induced by inflammatory stimuli through activation of NF-Kb [157].
Should be:
LCN2 is abundantly expressed in WAT and is induced by inflammatory stimuli by activating NF-Kb [157]. Revised according to reviewer's suggestion
Instead of:
It is one of the novel adipocytokine that influences glucose metabolism serving as predictor for diabetes [67], and has anti-inflammatory [68], anti-atherosclerotic, and cardiovascular protective effects [69].
Should be:
It is one of the novel adipocytokines that influences glucose metabolism serving as a predictor for type 2 diabetes [67], and has anti-inflammatory [68], anti-atherosclerotic, and cardiovascular protective effects [69]. Revised according to reviewer's suggestion
Instead of:
The correlation between chemerin with BMD has been consistently negative [7-9] whereas no convincing evidence has been found on the association between visfatin [10, 11] and Lipocalin 2 [12, 13] with BMD.
Should be:
The correlation between chemerin with BMD has been consistently negative [7-9], whereas no convincing evidence has been found on the association between visfatin [10, 11] and lipocalin 2 [12, 13] with BMD. Revised according to reviewer's suggestion
Instead of:
NAMPT /visfatin is involved in energy metabolism and also known to have pro- inflammatory properties [51], thus, may be useful as potential biomarker for obesity, insulin resistance and Type 2 diabetes (T2D) status [52].
Should be:
NAMPT/visfatin is involved in energy metabolism and is also known to have pro-inflammatory properties [51]; thus, it may be helpful as a potential biomarker for obesity, insulin resistance and type 2 diabetes (T2D) status [52]. Revised according to reviewer's suggestion
Instead of:
In 2007, Bozaoglu et al. reported the level of chemerin and its receptor, chemokine-like receptor 1 (CMKLR1), were significantly high in individuals with obesity and Type 2 diabetes [59].
Should be:
In 2007, Bozaoglu et al. reported that the level of chemerin and its receptor, chemokine-like receptor 1 (CMKLR1), were significantly high in individuals with obesity and type 2 diabetes [59]. Revised according to reviewer's suggestion
Instead of:
It is also known as Neutrophil gelatinase-associated lipocalin (NGAL) [79].
Should be:
It is also known as Neutrophil gelatinase-associated lipocalin (NGAL) [79]. Revised according to reviewer's suggestion
Instead of:
In 2006 Oh-1 and colleagues identified a novel anorexigenic adipokine nesfatin also known as non-esterified fatty acid/nucleobindin2 (NEFA/NUCB2) produced in the hypothalamus and released as a fragment, nesfatin-1 into the circulaion.
Should be:
In 2006 Oh and colleagues identified a novel anorexigenic adipokine nesfatin, also known as non-esterified fatty acid/nucleobindin2 (NEFA/NUCB2) produced in the hypothalamus and released as a fragment, nesfatin-1 into the circulation. Revised according to reviewer's suggestion
Instead of:
A similar study in ovariectomized rats showed that administration of nesfatin led to a significant increase in serum osteocalcin and BAP andTvBMD and TbvBMD by pQCT in the metaphysis of long bones compared to the control group [111].
Should be:
A similar study in ovariectomized rats showed that administration of nesfatin-1 led to a significant increase in serum osteocalcin and BAP and TvBMD and TbvBMD by pQCT in the metaphysis of long bones compared to the control group [111]. Revised according to reviewer's suggestion
Instead of:
In contrast, a more recent study demonstrated that Apelin was down-regulated in rats with ovariectomy-induced osteoporosis and treatment with Aplein-12 restored bone mass and microstructure [121].
Should be:
In contrast, a more recent study demonstrated that apelin was down-regulated in rats with ovariectomy-induced osteoporosis and treatment with apelin-12 restored bone mass and microstructure [121]. Revised according to reviewer's suggestion
Instead of:
There is evidence suggesting that it could potentially ameliorate vascular complications from hyperglycemia in patients with diabetes [3].
Should be:
Evidence suggests that it could potentially ameliorate vascular complications from hyperglycemia in patients with type 2 diabetes [3]. Revised according to reviewer's suggestion
Instead of:
Studies confirmed that extracellular signaling proteins including the adipocyte specific (PPARg) or myogenic osteoblast initiation (inhibition of PPARg or activation of Runx2/osterix) transcription factors can alter the differentiation of osteoblasts[136-138].
Should be:
Studies confirmed that extracellular signaling proteins, including the adipocyte-specific (PPARgamma) or myogenic osteoblast initiation (inhibition of PPARgamma or activation of Runx2/osterix) transcription factors, can alter the differentiation of osteoblasts [136-138]. Revised according to reviewer's suggestion
Instead of:
Chimerin, a novel adipokine with its receptor CMKLR1, have direct interaction with PPARc and promoted MSC adipogenesis [150].
Should be:
Chemerin, a novel adipokine with its receptor CMKLR1, have direct interaction with PPARc and promotes MSC adipogenesis [150]. Revised according to reviewer's suggestion
Round 2
Reviewer 2 Report
Narrative reviews, according to the hierarchy of evidence, are at the bottom of the pyramid, there is a high risk of bias due to their subjectivity and lack of methodology, I advise against their publication in their current form, I recommend the authors to carry out a systematic review.
The authors do not comply with the guidelines of the journal in reference to citations in the text and bibliographic references section.